# Oil Spill Detection Using Machine Learning and Infrared Images

**Thomas De Kerf ***[ID]**, Jona Gladines** [ID]**, Seppe Sels** [ID] **and Steve Vanlanduit**

Op3Mech, Faculty of Applied Engineering, University of Antwerp, Groenenborgerlaan 171,
2020 Antwerp, Belgium; jona.gladines@uantwerpen.be (J.G.); seppe.sels@uantwerpen.be (S.S.);
steve.vanlanduit@uantwerpen.be (S.V.)
* Correspondence: thomas.dekerf@uantwerpen.be

**Abstract:** The detection of oil spills in water is a frequently researched area, but most of the research has been based on very large patches of crude oil on offshore areas. We present a novel framework for detecting oil spills inside a port environment, while using unmanned areal vehicles (UAV) and a thermal infrared (IR) camera. This framework is split into a training part and an operational part. In the training part, we present a process for automatically annotating RGB images and matching them with the IR images in order to create a dataset. The infrared imaging camera is crucial to be able to detect oil spills during nighttime. This dataset is then used to train on a convolutional neural network (CNN). Seven different CNN segmentation architectures and eight different feature extractors are tested in order to find the best suited combination for this task. In the operational part, we propose a method to have a real-time, onboard UAV oil spill detection using the pre-trained network and a low power interference device. A controlled experiment in the port of Antwerp showed that we are able to achieve an accuracy of 89% while only using the IR camera.

**Keywords:** oil spill detection; machine learning; infrared imaging; image segmentation; drone imaging

## 1. Introduction

Large oil spills, such as the Gulf War oil spill (1991), The Kolva River spill, and more recent the Deepwater Horizon accident are disastrous for the environment. This impact has been studied in [1,2]. These oil spill events are featured prominently in the media, but most oil spills are smaller (<700 tonnes of oil) and they occur in, or near, ports. According to the latest oil tanker spill statistics report [3], 66% of the registered oil spill are medium size (7–700 tonnes) and 53% of those oil spills occur inside a port. The current method of identification is based on coincidence. When a port authority inspector notices an oil spill, they notify their superior and the cleaning company is notified. During nighttime, the odds of detecting an oil spill are lowered significantly, since not every part of the water is illuminated inside of the port. This evaluation method that is based on coincidence can cause a large time gap between the oil spill incident and start of the cleaning procedure. In [4], it is shown that oil slicks can achieve speeds of 0.4 to 0.75 cm/s for port conditions. Hence, to within one hour, the oil spill could be more than 2 km further. Therefore, a fast identification (less than 30 min) is important to achieve following advantages:

- Less harmful environmental damage. This is because of the fact that the cleanup is faster.
- The oil can be contained faster. This will make the cleanup process more efficient.
- Less impact on the rest of the operations inside the port, which results in less economical damage.
- In certain situations, it will lead to a more clear indication of the polluter.

The detection of oil spills is a widely researched subject. The Deepwater Horizon disaster was a catalyst for a high amount of research activities, see [5–9]. Oil tanker accidents can also cause a severe environmental impact. Research into detection strategies and mitigation methods, on a confined marine basin, such as the Baltic Sea or the Mediterranean Sea, can be found in [10–12]. Regarding the use of remote sensing technologies in a port environment for detecting oil spills, several factors are specific to said port environment. These include the type of oil (crude oil vs refined oil), the thickness of the oil (thin oil sheen's in ports vs thick patches in ocean environments), and the size of the oil patch.

Because specific research on oil spills in ports with drone technology is sparse, we propose a drone based hybrid RGB/thermal infrared solution. While using thermal infrared imaging, we are able to detect the oil spill on water, because oil will absorb light in the visible region and re-radiates a portion of that light in the thermal infrared spectrum [13,14]. Because oil has a greater infrared emissivity than water, it will appear hot as compared to water in the daytime. During nighttime, this effect will reverse.

This study proposes an automated detection of oil on water using only thermal infrared images utilizing a CNN (Convolutional Neural Network). A dual setup is used to create a large enough dataset, comprising of a RGB camera that records images in the visible spectrum and an Infrared camera that records images in the thermal infrared spectrum (7 μm–14 μm). The RGB images are then annotated while using thresholding algorithms and they serve as a ground truth mask as input for the CNN, together with the infrared images. Once this network is trained, it will enable frequent inspections at low cost. This novel approach to UAV oil spill detection combines state of the art CNN segmentation architectures and the use of thermal infrared imaging. This is demonstrated in a test setup, where we are able to successfully detect oil with an accuracy of 89%. With this same technique, we could also detect oil during nighttime.

The remainder of this article is structured, as follows. In Section 2, we present an overview of the current technologies used to detect oil spills. In Section 3, the detailed methodology of our framework is presented. Section 4 contains the experimental setup and accompanying results and, in Section 5, we draw conclusions and present possibilities for future work.

## 2. Literature

We start from the work of Fingas et al. [13], where an overview is given for multiple remote sensing technologies used in order to detect crude ocean oil spills. These technologies are well suited to be mounted inside an airplane or satellite to inspect wide areas, but, for operational UAV use in a port, there are different aspects that have to be accounted for. These decisive factors are: cost, availability, and applicability for UAV.

Radar is widely used to detect oils spills while using planes. Synthethic aperture radar or SAR has shown to be effective in differentiating between oil, algae, plankton, and other false positives [6,15,16]. However, light weight SAR that can be mounted under a small UAV is still under development [17], and it is not yet tested to detect oil spills.

Another detection method is based on fluorescence. When using fluorescence, a light source (usually an ultraviolet (UV) laser) is pointed at the target, as shown in [18]. An UAV setup is discussed in [19]. That research showed promising results, with a 1000mW laser during nighttime. However, during daytime, a much more powerful laser should be used. Therefore, this technology is not deemed to be suitable for daytime detection.

Detection in the UV spectra is also possible. Research [20] shows that water and an oil film have a different reflection coefficient in the UV wavelength range. A recent paper by [21] demonstrated that the sensitivity of UV reflectance images (365 nm) for oil spill detection is much higher when compared to the visible wavelength band. Unfortunately, UV images are also subject to interference, such as sun glints, wind slicks, and biogenic material. There are algorithms, [22], which can reduce interference. Detection during nighttime is not possible without an external source of light.

Oil spills can be detected while using thresholding techniques in the Near Infrared (NIR) spectrum (750–1000 nm), as shown in [23], but this is prone to false negatives. In the Short Wave Infrared (SWIR) band (1000–1700 nm), crude oil has a different reflectance spectrum when compared to water [24]. The use in low light conditions is tested in [25], and it has been proven to be successful. The high cost and the limited commercial availability is the downside of this solution.

Several satellites provide reflection data in the visible wavelength range, for instance, the Sea-viewing wide Field-of-Wiew Sensor (SeaWiFS) radiometer of the SeaStar satellite. The advantage of the recorded visible range images is that multi-spectral data are available (e.g., RGB colored images). Hence, it is possible to combine information in the separate wavelength bands in order to detect oil spills [26]. In [27], it is shown that it is possible to differentiate between water, oil slicks, and interference, such as algae. In [28] a solution is presented for automated detection while using object detection based on RGB cameras mounted on a UAV. The implementation is successful, but, using only this technology, it is not possible to detect oil spills during nighttime.

When considering all of the aforementioned technologies and the specific requirements, we found that a solution using the thermal infrared region is the best suited. Oil will absorb light in the visible region and it re-radiates a portion of that light in the thermal infrared spectrum [13,14]. Oil shows greater infrared emissivity than water, so it will appear hot when compared to water. However, thin sheens of oil cause a destructive interference of the Infrared wavefronts and therefore appear cool. Hence, during the daytime, both higher and lower water surface temperature can indicate an oil spill. During nighttime the effect will reverse, oil will appear cool on less cool water.

The main advantages of using an IR camera technology are the possibility to detect during nighttime, a robust detection with few false positives, low cost, and light weight. A novel framework is presented in the following Section in order to predict the size and location of the oil spill, while using the raw thermal images.

## 3. Methods

This Section describes the proposed framework that we used to detect oil spills. This entire framework can be divided into two separate processes:

- **Training process:**
  To train the CNN, we use both of the RGB images and the Infrared images. From the RGB images, the oil spill is segmented, so that we have a mask representing the oil in the RGB image. We then feed both the segmented RGB images and IR images to the neural network and start the training process. Figure 1 presents an overview of that process.
- **Operational Process:**
  Once the training is done, the trained CNN can be deployed while using an interference device. Interference devices are low cost, low power computers, and are highly optimized for parallel GPU computations, ideal for CNNs. These devices make it possible to segment the images in real-time. See Figure 2 for a step-by-step flowchart.

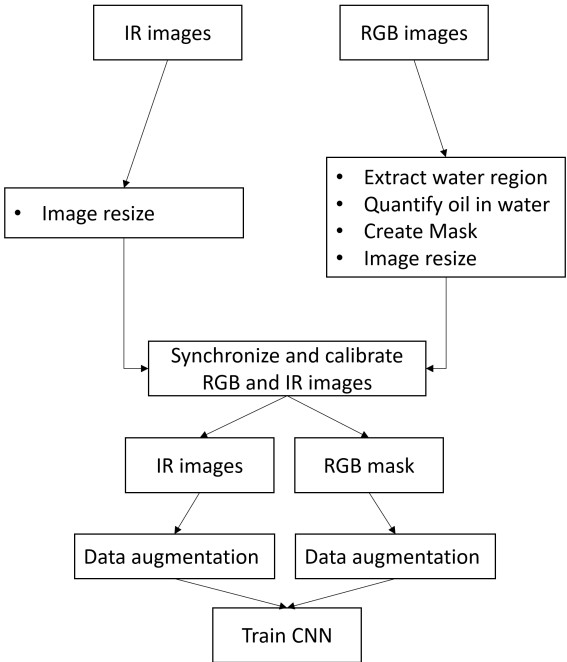

**Figure 1.** Flow chart of the training process.

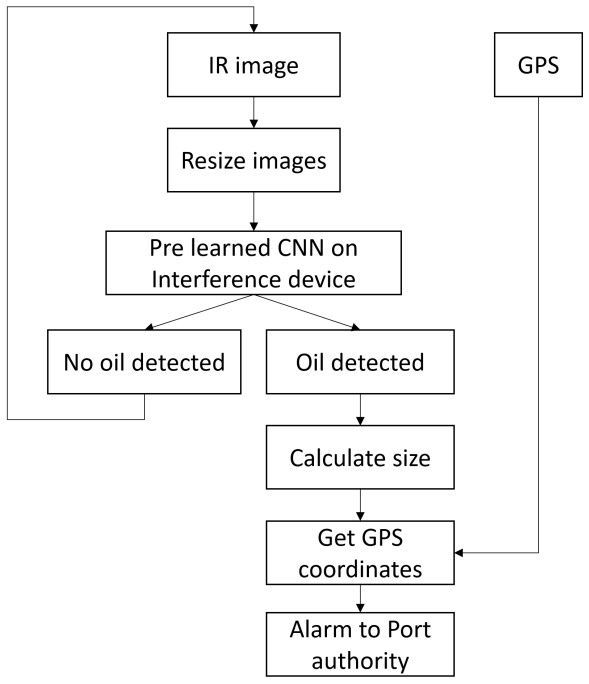

**Figure 2.** Flow chart of the operational process.

*3.1. Training Process*

In the following section, the training process is explained.

We start the training process with the raw IR and RGB images. For both cameras, we resize the images, from 3840 × 2160 pixels to 640 × 480, in order to reduce the training time. For the RGB images, we use thresholding algorithms to differentiate oil on the water surface. These steps are visualized in Figure 3. After these steps are done, we end up with a binary image mask. In this mask, there are two categories, oil and no oil.

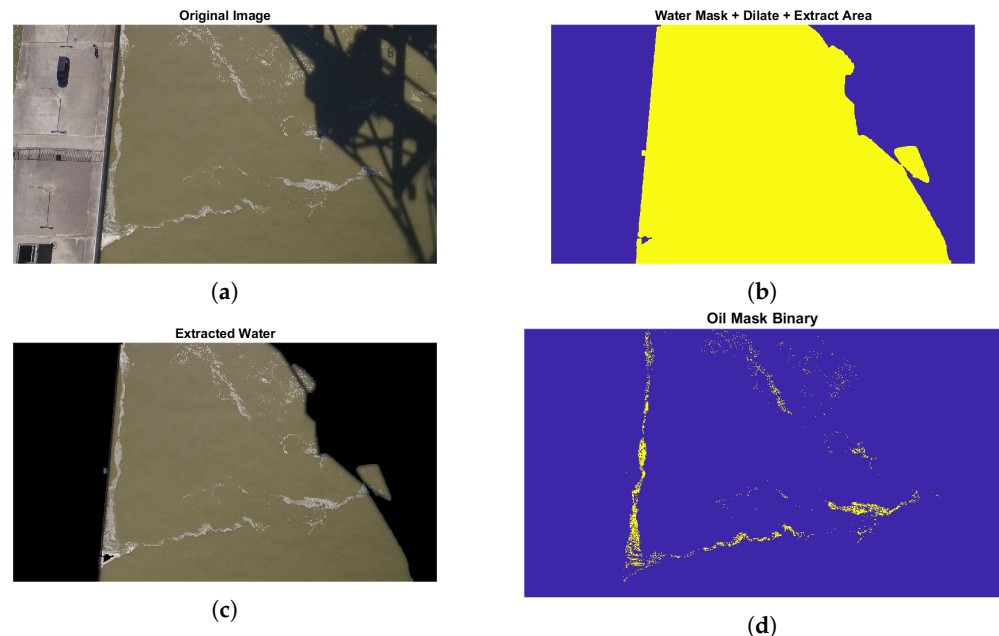

**Figure 3.** RGB Pipeline (**a**) Raw RGB image. (**b**) RGB image with the entire water/oil region extracted. (**c**) Raw RGB image with water/oil mask applied. (**d**) Oil extracted from (**c**).

When these steps are finished, we end up with an IR image and a binary image with the annotated oil. Before feeding them to a neural network, the IR images on the mask are artificially augmented. Data augmentation is a common technique to improve the robustness of a neural network. Using data augmentation, we artificially create new images to match real life conditions. In this paper, five augmentation techniques are implemented: horizontal and vertical flip, motion blurring, cropping, and affine transformation. These steps are implemented while using the popular python package Imgaug, see [29]. A visualization of the augmentation techniques is made visible in Figure 4.

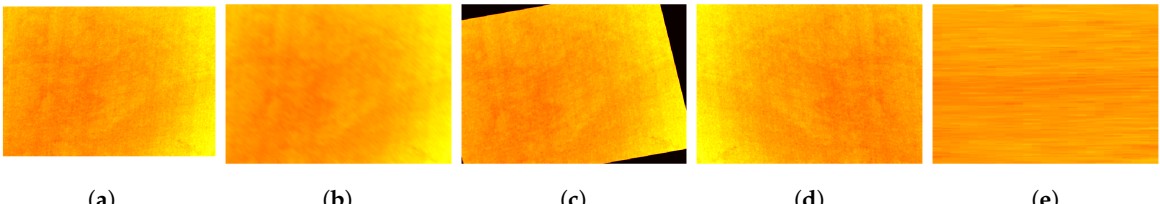

**Figure 4.** (**a**) Original thermal infrared image. (**b**) IR image with gaussian blur (**c**) IR image with affine transformation (**d**) IR image horizontally flipped (**e**) IR image cropped.

We then use these transformed images as input to train the neural network.

### 3.2. Training a Neural Network

Image segmentation is a computer vision technique, where every pixel is labeled into a predefined category [30] The result is an image that is segmented into several categories. This creates a simplification of the image; it makes it easier to visualize certain sectors with identical labels. Many architectures exist to implement image segmentation; the next Section presents the following well known architectures.

#### 3.2.1. Architectures

The following image segmentation architectures are evaluated (Unet [31], PSP [32], Segnet [33], and FCN [34]). Each of these architectures work in a similar fashion: first, the image is deconstructed (encoder) with convolutional layers in order to create a low dimensional tensor and, afterwards,

the image will be reconstructed (decoder) in order to display the segmented image. For the encoder part of the network, we can use a technique called transfer learning. Using transfer learning, we are able to take common neural networks that perform very well on image classification tasks and retrain only the top layers, in order to work with our data. We will reference these classification networks as feature extractors. The feature extractors used in this article are: Resnet50 [35], Mobilenet [36,37], VGG [38], and all of them are trained on the Imagenet dataset [39].

### 3.2.2. Hyper-Parameters

The following optimizers were used to evaluate the different models: Adadelta [40], Adamax [41], Nadam [42], Adagrad [41], and RMSprop [43]. The weight initialization was done while using He normal [44].

The most common evaluation metric is the accuracy. However, since the oil region is sparse, this is not the best metric. For instance, when there is only a fraction of oil in the water, predicting the entire region as water would yield a high accuracy. A metric that is more suited to handle such a high class imbalance is the mean Intersection over Union (mIoU) or Jaccard index, see Equation (1). This metric can be defined as the area over overlap, divided by the area of union and this for each class. Figure 5 shows an example of the difference between accuracy and mIoU applied to oil spill detection.

$$IoU = \frac{|A \cap B|}{|A \cup B|} = \frac{|A \cap B|}{|A| + |B| - |A \cap B|} \tag{1}$$

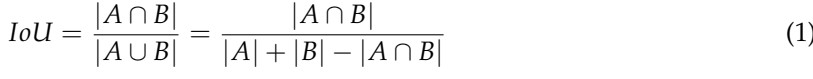

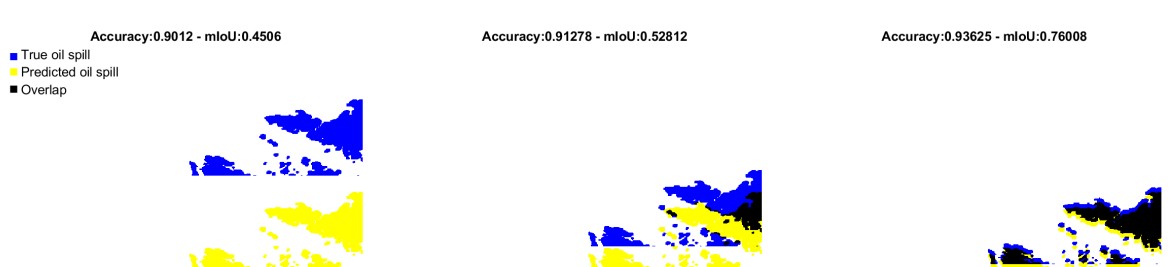

**Figure 5.** Calculated values of the accuracy and mean Intersection over Union (mIoU) for different cases. A high accuracy does not implicate a good overlap in our case.

### 3.3. Operational Process

Once we have the fully trained model, we can use this model to predict an oil spill with only the IR camera. This can be done in real time via an interference device. These interference device are small, low powered, and high performance devices that are optimized for parallel computing. The small size and low power consumption makes them the ideal choice for mounting on a UAV.

Every image will be resized and segmented by the CNN on board. If there is oil detected, then the device will gather all relevant information, such as size of the spill and location. It will then send an alarm to the port authority system, where the appropriate actions can be taken.

### 3.4. Experimental Setup

We created a controlled oil spill incident in order to test the our hypothesis and framework.

An area of four by four meters on the water is isolated by means of white adsorbing bands. Figure 6 shows the oil containing floating rectangle. This adsorbing material will soak up the oil, but not water. The same material is used by the port to contain oil spills. These bands were tied to the shore in order to fixate the oil spill. We were then able to contaminate the area with oil while filming with the fixed setup and the drone setup. Using these adsorbing materials, we could safely deposit oil into the water and be sure that is would be contained inside the square. A drone with two cameras (RGB and Thermal IR) hovered 30m above the square while we deposited three different kinds of

oil (HFO, DMA, and ULSFO), which are commonly used in the port of Antwerp. The drone used was a Acecore Neo with an Workswell Wiris 640 as thermal infrared camera and a Sony alpha 7r as RGB camera.

Due to the winds, and turbulence of the water, the oil did not stay in one place, but rather floated to the extent of the square, where it was absorbed by the white bands. When the deposited oil was absorbed by the bands, another oil deposition was done. Three separate flights were made with a flight time of 9 min. in order to ensure that enough training data were available. The oil deposition was done on multiple locations inside the square.

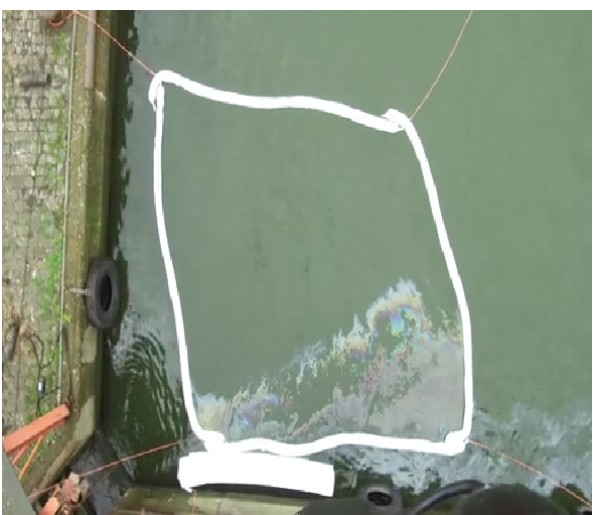

**Figure 6.** view from drone on test setup inside the port.

*3.5. Preprocessing*

The preprocessing steps, as described in Section 3.1, are performed on the captured data. However, that training process is slightly modified due to the nature of the setup. The shape of the rectangle was changed during the experiment because of the wind. Accordingly, a certain shape will correspond to a certain of amount of oil inside the rectangle. These are independent variables since no correlation exists between the size of the rectangle and the amount of oil. If we would feed these raw images to train the CNN, it is possible that the model could over-fit the data on the shape alone. Thus, achieving a high accuracy on the training data, but useless when evaluated with new data. In order to overcome this problem, several steps are performed to only extract the center part of the rectangle. Figures 7 and 8 illustrate the entire preprocessing process.

1.  Data compression of the image. The original footage was in 3840 × 2160 pixels large. This is compressed to 640 × 480 pixels in order to speed up the preprocessing

2.  Extraction of the image center and masking of the region outside the absorbing bands. The region of interest was the area where the oil was contained (in between the white absorbing strings). This region was identified using thresholding and edge detection image processing techniques. Figure 7b shows the result of this step.

3.  Transformation of the contained region presented in Figure 7b to a rectangular region. The region in the white absorbing bands was a deformed rectangle (because of the current). In order to transform the deformed region back to a rectangle, we first identified the corners and then applied a transformation matrix on the image (see Figures 7c and 8c, where this transformation was applied both on the binarized RGB image and on the infrared image).

4.  Estimation of the amount of oil inside of area. Using a thresholding algorithm, it was possible to estimate the amount of oil inside the area (thresholding means that we classify the pixels in an image based on the intensity value or color value of the pixel). This was possible in an accurate

way, because the conditions were optimal, and there was no direct sunlight or other disturbances on the image. This estimation will serve as a ground truth image. We save the resulting binary (oil = 1, no oil = 0) with the same name as the corresponding IR image. The result of applying a thresholding algorithm can be seen in Figure 7c.

In the end, we end up with an dataset of IR images and the correspondence ground truth masks. These training data were split into 70/20/10 train/validate/test categories.

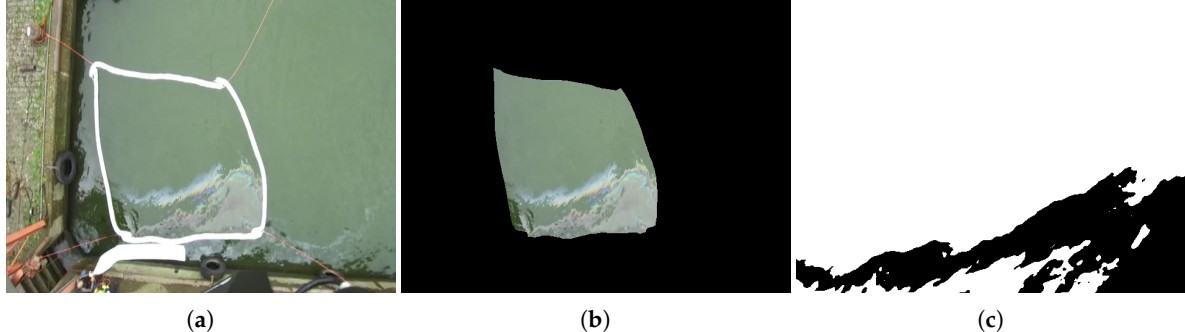

|            |            |            |
|:----------:|:----------:|:----------:|
| (**a**)    | (**b**)    | (**c**)    |

**Figure 7.** RGB Pipeline (**a**) Original RGB image. (**b**) RGB image with center extracted (**c**) Final mask, white color represents water, black represents oil.

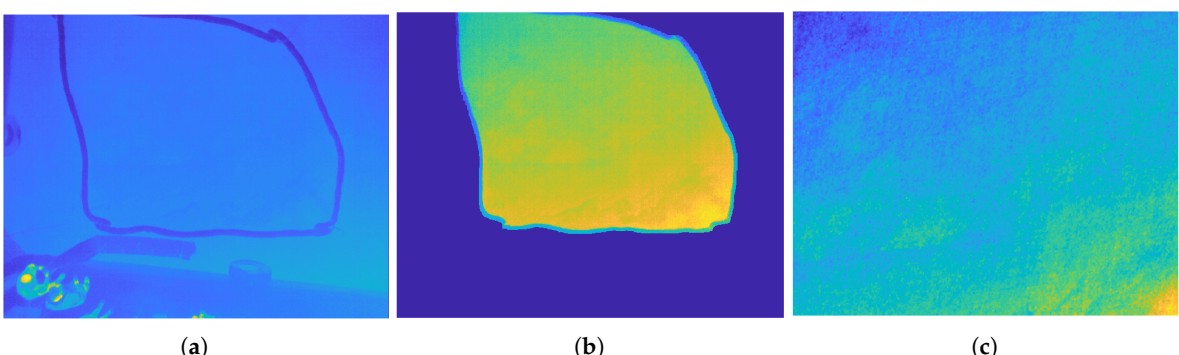

|            |            |            |
|:----------:|:----------:|:----------:|
| (**a**)    | (**b**)    | (**c**)    |

**Figure 8.** IR pipeline (**a**) Original thermal infrared image. (**b**) IR image with center extracted (**c**) IR image stretched to match size of the RGB mask.

### 3.6. Post Processing

All of the training was done on a 64bit PC with intel i9-7900x CPU (Santa Clara, CA, USA) and 2 Nvidia GTX 1080Ti GPUS (Santa Clara, CA, USA), each with 11 GB ram.The training is stopped when the network fails to improve for five epochs in a row. This early stopping feature maximizes the operational use of the training PC. The algorithms used and discussed in this paper are implemented with the popular deep learning Keras framework. Keras provides a high level programming environment in python and uses tensorflow as backend.

### 3.7. Nighttime Detection

During nighttime, it is not possible to use the RGB camera to segment the image into oil and non-oil categories. However, it is possible to use this camera in the transition from day to night or vice versa. These early or late hours have the same temperature conditions as during the night, thus enabling us to create a large database of comparable environmental conditions.

## 4. Results

In Table 1, the results of the neural network with different combinations of the above described parameters are shown. From the Table, we can conclude that the use of 'Mobilenet' as feature extractor, the results are remarkably better than the other feature extractors. From the results, we can conclude that the choice of the feature extractor has greater impact than the choice of network architecture. The!choice of the optimizer is another important factor. We observed that some optimizers are not able to fully converge on an acceptable accuracy.

The best combination of feature extractor, segmentation architecture, and optimizer is Mobilenet + FCN8 + RMSprop. The resulting network results in a mean IoU of 89% which is a very good value to achieve. In Figure 9, an IR image is processed by the neural network and a prediction is made. This is done on an image from the test set (which was not used to construct the neural network).

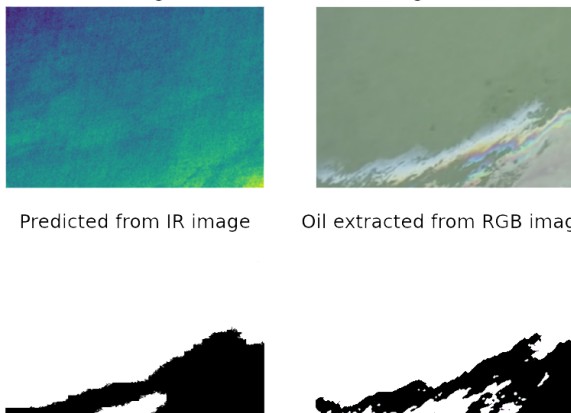

**Figure 9.** Oil classification result of the neural network (Mobilenet+FCN8+RMSprop). **Top**: infrared and RGB images. **Bottom-right**: mask from the RGB image using thresholding (see Section 3.1). **Bottom-left**: result of the trained neural network when applied to the infrared image (**top-left**).

**Table 1.** Results of the different architectures, feature extractors and optimizers. Note: only the result with the best optimizer is shown.

| Feature Extractor | Segmentation Architecture | Optimizer | Mean Iou | Val Mean Iou |
|---|---|---|---|---|
| MobileNet | FCN 8 | RMSprop | 0.89 | 0.89 |
| MobileNet | FCN 32 | Adamax | 0.88 | 0.88 |
| MobileNet | Unet | RMSprop | 0.87 | 0.87 |
| MobileNet | Segnet | RMSprop | 0.83 | 0.84 |
| ResNet 50 | Segnet | Adamax | 0.79 | 0.79 |
| ResNet 50 | Pspnet | RMSprop | 0.75 | 0.75 |
| Pspnet | Pspnet | Adamax | 0.65 | 0.65 |
| VGG | FCN 32 | Adadelta | 0.64 | 0.64 |
| FCN 32 | FNC 32 | Adadelta | 0.60 | 0.60 |
| FCN 8 | FNC 8 | Adadelta | 0.59 | 0.60 |
| VGG | Segnet | Adamax | 0.60 | 0.60 |
| VGG | Pspnet | RMSprop | 0.59 | 0.59 |
| Unet | Unet-mini | RMSprop | 0.55 | 0.55 |
| ResNet 50 | Unet | Adadelta | 0.49 | 0.49 |
| VGG | Unet | RMSprop | 0.47 | 0.48 |
| VGG | FCN 8 | Adadelta | 0.44 | 0.44 |
| Unet | Unet | RMSprop | 0.41 | 0.41 |
| Segnet | Segnet | RMSprop | 0.38 | 0.38 |

## 5. Conclusions

This research proposes a new approach for oil spill detection inside a port area while using an UAV and IR camera. Implementing this solution can increase the detection rate and decrease the overall cleaning costs of an oil spill. We tested our hypothesis and framework on a deliberately created oil spill under controlled circumstances. Using the Mobilenet Feature extractor and an FCN network architecture, we are able to achieve an mIoU of 89% from the dataset creating during the experiment. Furthermore, our framework can be implemented realtime and onboard a UAV. This allows for minimal human interaction during operation.

This method is primarily used in order to detect smaller oil spills that might go unnoticed by the port authority. This early-stage detection is crucial in aiding the efficient cleaning of the oil spill. During the test environment, we had a field of view of 31.9 m by 42.1 m, and were able to detect oil spills that fell within the field of view.

Further improvements could be investigated, such as: other camera technologies (SWIR (Short Wave Infrared) or Hyperspectral imaging), but also more advances in RGB preprocessing techniques could be implemented.

**Author Contributions:** Conceptualization, T.D.K. and J.G.; methodology, T.D.K., S.S. and J.G.; software, T.D.K.; validation, T.D.K.; formal analysis, T.D.K.; investigation, J.G. and T.D.K.; resources, J.G. and T.D.K.; data curation, T.D.K.; writing—original draft preparation, T.D.K.; writing—review and editing, T.D.K. and S.V.; visualization, T.D.K. and S.V.; supervision, S.V. and S.S.; project administration, S.V.; funding acquisition, S.V. All authors have read and agreed to the published version of the manuscript.

**Funding:** This research was funded by VLAIO TETRA project AutoDrone.

**Acknowledgments:** We thank to the Port Of Antwerp for the financial support and for enabling the in-situ measurements. We thank HAVIQ for performing the drone flights.

**Conflicts of Interest:** The authors declare no conflict of interest.

## Abbreviations

The following abbreviations are used in this manuscript:

| | |
|---|---|
| MDPI | Multidisciplinary Digital Publishing Institute |
| DOAJ | Directory of open access journals |
| UAV | Unmanned Aereal Vehicle |
| UV | Ultra Violet |
| mIoU | mean Intersection over Union |
| NIR | Near Infrared |
| SWIR | Short Wave Infrared |
| CNN | Convolutional Neural Network |

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
