# Peer review of "Oil Spill Detection Using Machine Learning and Infrared Images"

_remotesensing, doi:10.3390/rs12244090_

Round 1

Reviewer 1 Report

Dear authors

I liked to read your paper, having only very minor comments to convey.

  1. The movement and eventual mitigation of medium to small oil spills have also been addressed by European researchers working in the NEREIDs and Sea4All projects, funded by the EU. The fist half of the Introduction is, therefore, too harshly written bearing in mind the work developed in the Eastern Mediterranean, Sicily and Baltic Sea, to mention three common bottlenecks for shipping with some frequent spills.
  2. There is no reference to the limits in size in oil spills that will render this method less effective. I believe that oil spills can become so large (even close to the shore and harbours) that you will not need to use your method whatsoever.

I attached an annotated .pdf with these comments to this review. I feel these are important aspects that need to be addressed in a moderate/minor revision.

Author Response

Reviewer Comments:

Point 1:

The movement and eventual mitigation of medium to small oil spills have also been addressed by European researchers working in the NEREIDs and Sea4All projects, funded by the EU. The fist half of the Introduction is, therefore, too harshly written bearing in mind the work developed in the Eastern Mediterranean, Sicily and Baltic Sea, to mention three common bottlenecks for shipping with some frequent spills.

Response 1:
Thank you for pointing this out, the reviewer is correct in suggesting the European projects.

I have included the following paragraph in the paper to address the reviewers comment:
The detection of oil spills is a widely researched subject. The Deepwater Horizon disaster was a catalyst for a high amount of research activities, see \cite{Hu2018,Leifer2012,White2016,Garcia-Pineda2013,Xing2015}. Oil tanker accidents can also cause a severe environmental impact. Research into detection strategies and mitigation methods, on a confined marine basin such as the Baltic Sea or the Mediterranean Sea, can be found in \cite{Alves2014, Alves2015,Andrejev2011}.
As for using remote sensing technologies in a port environment to detect oil spills, several factors are specific to said port environment. These include the type of oil (crude oil vs refined oil), the thickness of the oil (thin oil sheen's in ports vs thick patches in ocean environments), and the size of the oil patch

Point 2:

There is no reference to the limits in size in oil spills that will render this method less effective. I believe that oil spills can become so large (even close to the shore and harbors) that you will not need to use your method whatsoever.

Response 2:

The reviewer is correct that the maximum size of the oil spill detection is not mentioned in the paper. The reviewer is also correct that when the oil spill is very large, our method will not aid in the detection of the oil spill. Large oil spills will not go unnoticed very long in a busy port environment. That is why this research mainly focuses on smaller oil spills that can go undetected by

To address the concerns of the reviewer following paragraph is added to the conclusion section:
This method is primarily used to detect smaller oil spills that might go unnoticed by the port authority. This early-stage detection is crucial to aid the efficient cleaning of the oil spill. During the test environment we had a field of view of 31.9m by 42.1m and were able to detect oil spills that fell within the field of view.

Reviewer 2 Report

Review of  “ oil spill detection using machine learning and infrared images”, by De Kerf et al.

The authors use an RGB camera, imaging a controlled oil slick under ideal conditions, to provide a training set for neural network processing images of the same spill with a thermal IR sensor. The paper is properly submitted as a technical note as there is little new research here. Using different combinations of multi-spectral scanners producing spill images processed by neural networks has been employed with variable success for over two decades. However, the authors show a commendable job of meeting the technical difficulties involved (e.g. right choice of metric, limiting the output to yes/no oil over a set threshold, and employing a neural network method that allows some pattern recognition instead of just pixel-by-pixel rules), thus outlining a practical approach that Port authorities might use to detect illegal oil releases at night. Therefore, the reviewer recommends publication with a couple of slight modifications.

  • The authors, of necessity, calibrate the thermal IR during the day (no RGB images at night). However, oil thermal signals at night will likely be different. The authors should discuss how to meet this challenge.
  • The authors use a single oil product. A short discussion of how to adapt their method to different products and different states of weathering should be included. It looks like there may have been some relevant material that may have been accidentally deleted (see grammatical note below).
  •  

As a grammatical note, the sentence on page 2 that begins “Due to difference…} is incomplete.

Author Response

Reviewer Comments:

Point 1:
The authors, of necessity, calibrate the thermal IR during the day (no RGB images at night). However, oil thermal signals at night will likely be different. The authors should discuss how to meet this challenge.

Response 1:
Thank you for raising this question. The detection during nighttime is indeed a very challenging problem. Since we detect using temperature differences and not reflection of sunlight, the detection should also work during nighttime. The difference would be that the temperature is lower during nighttime compared to daytime. But this problem can be mended by performing flight in the transition from night to day or vice versa. Then we can still use the RGB camera to create a representative nighttime dataset.

I have included the following section (3.7) in the paper to address the reviewers comment:
During nighttime it is not possible to use the RGB camera to segment the image into oil and non oil categories. However, it is possible to use this camera in the transition from day to night or vice versa. These early or late hours have the same temperature conditions as during the night, thus enabling us to create a large database of comparable environmental conditions.

Point 2:
The authors use a single oil product. A short discussion of how to adapt their method to different products and different states of weathering should be included.

Response 2:
During the test, we used three different oil products that are commonly found in the port of Antwerp. I clarified this by adding the following text in section 3.4:

A drone with two cameras (RGB and Thermal IR) hovered 30m above the square while we deposited three different kinds of oil (HFO, DMA, and ULSFO) that are commonly used in the port of Antwerp.

To account for a change in weather conditions, the authors note that a sufficiently large dataset with different weather conditions should suffice to still have an accurate detection during all weather conditions.

Point 3:
As a grammatical note, the sentence on page 2 that begins “Due to difference…} is incomplete.

Response 3:
The sentence has been replaced by:
As for using remote sensing technologies in a port environment to detect oil spills, several factors are specific to said port environment. These include the type of oil (crude oil vs refined oil), the thickness of the oil (thin oil sheen's in ports vs thick patches in ocean environments) and the size of the oil patch